# Learner Experience of an Online Co-Learning Model to Support Mental Health during the COVID-19 Pandemic: A Qualitative Study

**DOI:** 10.3390/ijerph20032498

**Published:** 2023-01-31

**Authors:** Catherine Briand, Regis Hakin, Julio Macario de Medeiros, Francesca Luconi, Brigitte Vachon, Marie-Josée Drolet, Antoine Boivin, Catherine Vallée, Sarah Montminy

**Affiliations:** 1Research Center, Montreal University Institute of Mental Health, Montreal, QC H1N 3M5, Canada; 2Department of Occupational Therapy, University of Quebec at Trois-Rivières, Trois-Rivières, QC G8Z 4M3, Canada; 3Office for Continuing Professional Development, Faculty of Medicine and Health Sciences, McGill University, Montreal, QC H3G 2M1, Canada; 4School of Rehabilitation, Faculty of Medecine, University of Montreal, Montreal, QC H3N 1X7, Canada; 5Department of Family Medicine, Research Centre of University of Montreal Hospital Center, Montreal, QC H2X 0A9, Canada; 6Department of Rehabilitation, Faculty of Medicine, Laval University, Quebec City, QC G1V 0A6, Canada; 7VITAM Research Centre on Sustainable Health, Laval University, Quebec City, QC G1V 0A6, Canada

**Keywords:** mental health, mental health promotion, mental health prevention, psychological well-being, recovery, Recovery College, online education, adult education program, co-learning, COVID-19, self-care, self-determination, empowerment, stigmatization

## Abstract

The COVID-19 pandemic has had a negative impact on the mental health of the population such as increased levels of anxiety, psychological distress, isolation, etc. Access to mental health services has been limited due to the “overflow” of demands. The Recovery College (RC) model, an education-based approach, has addressed this challenge and provided online well-being and mental health courses to at-risk populations. The RC model proposes a co-learning space in an adult education program where learners from diverse backgrounds collectively learn and empower themselves to better address psychological well-being and mental health issues. The aim of this study was to document the experience of learners who participated in online RC courses during the COVID-19 pandemic and the perceived impact of these courses on their mental health. A qualitative interpretative descriptive study design was employed, and Miles and Huberman’s stepwise content analysis method was used to mine the data for themes. Fourteen structured online interviews were conducted with a sample representative of the diversity of learners. Five categories of themes emerged: (1) updating and validating your mental health knowledge, (2) taking care of yourself and your mental health, (3) improving and modifying your behaviors and practices, (4) changing how you look at yourself and others, and (5) interacting and connecting with others. Results suggest that online RC courses can be an effective strategy for supporting individual self-regulation and empowerment, breaking social isolation, and reducing the effects of stress in times of social confinement measures and limited access to care.

## 1. Introduction

Societies today face significant mental health issues that have been exacerbated by the COVID-19 (COVID-19) pandemic [1,2,3,4]. While everyone has been affected to some degree, large segments of the population (e.g., women, students, gender-diverse individuals, health care workers, and people with mental or chronic illness) have had to deal with greater risks to their mental health [5,6,7,8,9,10]. High levels of anxiety and psychological distress have been documented in many at-risk groups and in the general population [1,11]. Moreover, access to mental health services was overburdened since the demand was expanding rapidly. To reduce these risks and reinforce protective factors, innovative public health strategies geared to prevention and support have been implemented [1]. It is in this context that an online education initiative based on the Recovery College (RC) model was developed and evaluated in Québec as a response to the impact of COVID-19 on mental health [12,13,14]. 

Initiated in the UK and then applied in more than twenty countries, the RC model is predicated on a community education-based approach to give everyone free access to well-being and mental health workshops/courses [15,16]. It has been one of the fastest-growing mental health learning environment models in the world in the past five years [17]. Its distinguishing features include the following: (a) the creation of a co-learning space that recognizes all types of knowledge (theoretical, clinical, practical, experiential); (b) the co-development and co-delivery of courses by a dyad of trainers (peer and practitioner trainers); (c) the hybridization and cross-pollination of knowledge through innovative, participative, and active pedagogical methods; (d) a social diversity of learners and trainers from different backgrounds (e.g., education and health care providers, university students, people in vulnerable situations, family/peer caregivers, workers, managers, retirees/seniors, citizens); and (e) the promotion of horizontal and egalitarian social relationships that are benevolent and free of judgment and where people are encouraged to express their opinions, speak freely, and share power [18,19]. The model’s foundational principles rest on the essential equality of knowledge and human beings, on the action of co-learning and co-creation that allows the egalitarian participation of every learner, and on the experience of prejudice-free egalitarian relationships in a safe and inclusive space [18,20]. 

The Recovery College model has been the subject of many studies around the world. A synthetic review of 31 of these confirmed the positive effects of the model on numerous variables [21]. The RC curriculum and courses has been shown to improve the following: (a) mental health literacy (knowledge of mental health); (b) ability to care for oneself and use self-regulation and self-management strategies; (c) empowerment and the ability to make choices for oneself; (d) self-esteem and sense of competence; (e) psychological well-being and anxiety levels; (f) recovery and social isolation for people living with mental disorder; and (g) attitudes and behaviors of openness to others, including the ethical management of prejudice and reduction in stigmatization [13,14,16,22,23,24,25,26,27,28,29]. Furthermore, learners who take RC courses expand their network of contacts and resources and their ability to find help, while reducing their use of specialized care services [22,30,31]. The significant outcomes found in these studies were for RC curriculum and courses offered in person over a variable average amount of time (from a few hours to several days), with the possibility of attending more than one course (high effect size between 0.75–0.86 [Cohen’s d] [27,28]. In the COVID-19 situation where the mental health demand was high, free courses were instead offered online, of short duration, and with the possibility of only one course per person. Our hypothesis is that it has reached a more heterogeneous population than traditional mental health services and therefore provides a new type of mental health support. To date, no study has confirmed these results for a single, short, online CR course.

The Centre d’apprentissage Santé et Rétablissement (CASR), the only French-language RC in Canada, began offering an RC curriculum and courses to the French-speaking population of Québec in the fall of 2019. In the fall of 2020, because of the public health measures imposed to contain COVID-19, the CASR launched a curriculum of several shortened online RC courses (three two-hour sessions for a total of six hours per course) intended to increase course accessibility. The preliminary results of a demonstration project [12] funded by the Canadian Institutes of Health Research to assess the program’s impact were promising and consistent with findings from efficacy studies of longer, in-person RC courses [13,14]. To gain a better understanding of the effects of this shortened online RC curriculum offered by the CASR during the COVID-19 pandemic, the aim of this qualitative study is to document learner experience and perceived impact on mental health. Results would help to solidify the understanding of the impact, optimize and scale up educational approaches to mental health and thus provide a better response to the needs of the population.

## 2. Materials and Methods

A simple interpretative descriptive qualitative design was used to document the learner experience of RC courses and its perceived impact on mental health [32]. The design was chosen because we wanted each learner’s experience to enrich the explanation of the phenomenon under study. In addition, we hoped that the results of this qualitative study would inform the choice of measurement tools to be used in a subsequent experimental study in order to improve the evaluation process. Finally, in the context of collaborative evaluative research with decision-makers, the use of a qualitative design enhances their ability to gain a better understanding of learner needs [32]. 

### 2.1. Recruitment Process

Everyone who attended at least one of six courses offered by the CASR in the fall of 2020 was invited to participate in the study. Participation entailed completing questionnaires and sitting through an individual interview for 60 to 90 min. Of 201 learners, 44 signed a consent form to participate in the study and expressed an interest to be interviewed. 

As recommended in qualitative research and to ensure an appropriate representation of the lived experiences and diversity of CASR learners, a purposive sampling method was used with four criteria: courses attended, gender, age group, and learner profile can be seen in Table 1 below [32,33]. 

The project was approved by the research ethics committees of the Université du Québec à Trois-Rivières (#CER-20-270-07.01) and the Centre intégré universitaire de services sociaux et de santé de l’Est-de-l’Ile de Montréal (#MP-12-2021-2421). Participants provided written informed consent to take part in the study.

### 2.2. Data Collection

Of the 44 learners who expressed interest in being interviewed, 21 who met the criteria of the purposive sampling strategy were invited to be interviewed. The aim was to arrive at a final diversified sample of 12 to 15 learners. According to the theoretical saturation principle in qualitative research, 12 to 15 interviews are necessary to reach a saturation of new information [32]. In the end, 14 interviews were conducted (five learners declined, two cancelled). Table 2 presents the participants’ characteristics.

The interviews were conducted through the Zoom Videos Communication platform and were transcribed in their entirety for analysis. They were conducted following a structured interview protocol. The interview guide covered the learning experience at the CASR and learner expectations (Theme 1), understanding of the RC model and its action mechanisms (Theme 2), and the perceived and observed benefits of the courses (Theme 3).

### 2.3. Data Analysis

Data were analyzed using the stepwise qualitative content analysis method developed by Miles and Huberman [34]. This three-stage method serves the purpose of carrying out a subjective interpretation of textual data content through a process of systematic classification, coding, and theme identification. It is a highly rigorous method comprising three stages of coding laid out in detail in a codebook (themes with definitions). Additionally, its coding and analysis process is performed by multiple coders to ensure the validity of the themes that emerge to a good level of inter-coder agreement. The three stages of coding are: (1) an open-ended coding process to identify the ideas brought forward; (2) an axial coding process to group ideas into categories; and (3) a selective coding process to align with the RC model [34,35]. To ensure the quality of the analysis process, all transcripts were analyzed using the NVivo qualitative software tool. Additionally, the team used a mix of deductive (i.e., based on a theoretical framework of the documented effects of the RC model) and inductive (i.e., respecting the learners' experience) thematic analysis. To ensure the consistency and quality of the qualitative analyses across coders, a codebook was developed iteratively. 

The first two stages of the thematic analyses were performed by the first and second authors of this paper (CB and RH). The average inter-coder agreement was 82%. In an iterative process, inter-rater reliability was obtained by discussing the coded excerpts until a consensus was reached between the two coders. Then, the preliminary results were presented to the other three co-investigating authors of the project (BV, MJD and FL) to ensure the conceptual consistency of the resulting themes. After the themes and categories were adjusted based on the comments received from the co-investigators, the third stage of the thematic analysis was carried out by the first author (CB) and finally validated by all the co-authors. 

The use of a theoretical framework for analysis, a codebook, an iterative process of coding and analysis by multiple coders on all the material gathered ensures rigor, reproducibility, and transferability of results in qualitative research [32,34,35].

For replication purposes, the data that supports the findings of this study is available on request from the corresponding author. The data is not publicly available owing to privacy or ethical restrictions.

## 3. Results

Of the 14 learners interviewed, 11 were women. The average age of the learners was 44 years (range of 20 to 66). Learner profiles varied as described in Table 2. 

Five categories and 14 themes emerged from the qualitative thematic stepwise analysis. These are presented in Table 3 in the decreasing order of frequency with which they were mentioned by learners.

### 3.1. Updating and Validating Your Mental Health Knowledge

The category that emerged most frequently had to do with gaining new knowledge and validating existing knowledge, strategies, tools, and practices. The learners often mentioned that the course allowed them to gain new knowledge: 

(…) There were concepts I had never heard of. Of course, there are concepts you’re a little familiar with without being able to name them, but (…) it was still a concept that I never really grasped or that I never really paid attention to. (P10—Education professional)

(…) I was happy [to be] updated with new information (…) (P12—Peer worker, person with experiential knowledge)

The learners also mentioned how they updated and expanded their current knowledge, which sometimes made it easier to explain it to others:

(…) They [the trainers] taught me a lot of things. They made it easier for me to simplify some concepts better. (…) I was able to really understand and apply them. (…) (P4—Health professional) 

(…) There were certain things that I thought (…) [and that] weren’t quite right (…) now I have actual definitions for all of that. (P3—University student) 

(…) [it makes] it possible to share knowledge (…) at the end of the workshops, I even got the impression that I learned things or that it refreshed some things in my mind. (P9—Peer worker, person with experiential knowledge)

The learners also mentioned feeling validated when it came to the strategies, tools, and practices they were using:

(…) I realized that the things I learned through the course, those were things that, in general, I was applying without knowing it. It highlighted processes (…) that I was already bringing up on my own. (P1—Peer worker, person with experiential knowledge)

Finally, the learners mentioned that they wanted to keep learning and training: 

I learned a lot (…) from what others shared, from the information being presented, and from the different ways we were answering questions (…) it made me think a lot. It made me feel like learning more (…) (P11—Health professional)

(…) just one course isn’t enough (…) even if the original subject isn’t the same, there’s meeting other people, sharing, the experience (…) [I want] to keep learning. (P6—Peer worker, person with experiential knowledge)

### 3.2. Taking Care of Yourself and Your Mental Health

The second category to emerge most frequently concerned the perceived benefits related to the ability to take care of yourself and your mental health. The learners mentioned several times that they adopted new tools to take care of themselves and their mental health daily:

It gave me tools (…) to live better every day, because I was in a phase where I was very anxious. (P9—Peer worker, person with experiential knowledge)

(…) and the tool set (…) was like having a lot of things that we can apply to our lives or in our workplace. (P3—University student)

Every method we named which I noted down (…) I keep in mind what I can refer to (…) I try to apply them to my life (…) I can also help my clients, my patients with that. (P5—Health professional)

The learners also mentioned that they could better manage their stress and that they were more resilient and confident:

One day this week, I felt very emotional or frustrated. I thought back to the resiliency methods (…) I tried drawing from the suggestions that were made. (…) (P2—Citizen)

This course allowed me to [realize] that we can put things in perspective in life (…) that we can take in the stress in a caring mood, rather than fight it. (P4—Health professional)

The learners stated being more conscious of the importance of taking care of themselves and their mental health, which could result in feeling empowered in their daily lives: 

I am more conscious that I should make changes to my daily routine to improve my well-being (…) (P3—University student)

I learned that everyone, caregivers included, have to take care of themselves. I was saying that I worked a lot of hours each week. It reminded me that I need to find some kind of balance again. (P8 – Health professional)

It reminded me that I have power, that there are things I can do (…). (P9—Peer worker, person with experiential knowledge)

Finally, the learners felt more motivated to work on themselves and to seek help if needed: 

(…) it gave me a little boost to work on myself. (P10—Educational professional)

After two sessions, (…) I started looking at places where I could consult someone to talk about the problem I have interacting in groups (…) it inspired me to seek help. (P11—Health professional) 

### 3.3. Improving and Modifying Your Behaviors and Practices

The third category related to behavioral changes in daily practices to open up to others and fight stigmatization. Several times, learners mentioned that they wished that more value were ascribed to experiential knowledge (i.e., life experience): 

(…) the way I provide training… What I learned with you made me want to do it even more (…) differently. It changes some of the ways I was doing the training in that, while I covered the content that had to be covered, I should also emphasize lived experience. (P6—Peer worker, person with experiential knowledge)

(…) for my work (…) how to better integrate the experiences of the parents of my clients and the experiences of the clients themselves in (…) my clinical decisions (…) that’s an area where I could really improve (…) it encouraged me to really highlight all types of knowledge (…) (P11—Health professional)

The learners also reflected on the impact of stigmatization and what they could change through their own actions: 

(…) I think of young professionals who come out of school with the ambition of saving the world, knowing what’s good for [clients] and wanting to explain it to them (…) But it’s [more] like walking together, following the client (…) [the course] allows you to learn that, but from the inside. (…) open their minds (…) so that they see things from a different angle. (P14—Health manager)

I think it really made me become aware again [of] all the little things we might do every day that can stigmatize someone. We think of stigmatization as this big thing, but in fact, it’s all these little actions that everyone could be doing differently every day that would limit the big impact of stigmatization. It made me aware of the little things I could change or improve, ways of saying certain things differently to not hurt or stigmatize the other person. (P13—Health manager)

### 3.4. Changing How You Look at Yourself and Others

The fourth category concerned how the learners looked at themselves and others and how they changed this. They mentioned that the course led them to increase their self-awareness:

(…) What we learned isn’t so clear. You really have to think about it. (…) [those are] ways of looking at things (…) being less critical of yourself, more conscious of all the power we have in our lives (…) [those are] big lessons (…) I think that my professional life is where I’d try to promote that training. (…) I think there’s nothing like living it to (…) for it to make sense so that we remember later (…) (P14—Health manager)

I would say it’s the assessment. The assessment that I got to do of myself was very powerful (…) (P6—Peer worker, person with experiential knowledge)

The learners also mentioned that they modified and refined their understanding of otherness, that is, the fact that other persons were different from them:

[The course brought] other points of view. (P9—Peer worker, person with experiential knowledge)

Some of the sentences that other people said. By listening to their experience, there are certain things that stayed with me, that I still think about. It allows me another point of view. It’s another experience that I’ve lived. (P2—Citizen)

I’ll question myself, I’ll pay attention when I see someone who’s different from me (…) to put myself in that person’s shoes. I’ll tell myself: ‘they can also suffer from it’ (…) There’s a lot of personal questioning after this workshop (…) to become conscious again that how we look at others can influence how they feel. (P13—Health manager)

### 3.5. Interacting and Connecting with Others 

The fifth category regarded the space for interaction, socialization, and connecting with other people who participated in the course. The learners mentioned breaking their isolation and feeling less lonely during and after the course:

I felt better seeing that I wasn’t the only one living through that [speaking of the isolation because of COVID] (P5—Health professional)

It allowed me to socialize (…) to see people, to interact (…) to recognize them, see them again (…) it made me feel much better. Because during the pandemic, (…) I was alone much more than usual. (P9—Peer worker, person with experiential knowledge

The learners also mentioned that they expressed themselves more and that they regained confidence in themselves and in the value of their experiences and expertise:

It’s always a challenge for me to share my point of view. I tend to withdraw into myself (…) [the course] forced me to organize my thinking, then to express it, share it, make myself understood by the others. (P9—Peer worker, person with experiential knowledge)

(…) I really felt [that I] brought something valuable to this group because of my personal experience (…) that boosted my confidence and my desire to share those experiences there (…) (P11—Health professional)

## 4. Discussion

The purpose of this study was to document learner experience and perceived impact of the online RC courses offered by the CASR during the COVID-19 pandemic. The results show that, after the six hours of a course, learners perceived that they had improved their mental health knowledge and skills, their self-regulation and empowerment, their attitudes and behaviors of openness to others, and their confidence to speak about their mental health and to confide in others. Their level of awareness of the relevance of their own mental health and the mental health of others had increased as well. 

Our results align with international studies of RC curriculum and courses, despite focusing on a shortened online version of it. Earlier studies have indicated that participating in RC courses enabled learners to expand their knowledge of mental health and personal recovery [13,23,25,26,36], improve their ability to take care of themselves and to make time for it [13,16], and strengthen their ability to seek help [13,37]. They have shown also that learners gained a heightened sense of competence and hopefulness, broadened their knowledge, and adopted tools to take care of their mental health [13,14,16,38]. Several studies also reported improved personal well-being and stress management [14,16,28,39,40]. Learners in other studies have qualified the training sessions as unique occasions for creating connections with others and socializing in a welcoming, supportive, egalitarian, and inclusive environment [16,36,41]. Previous studies have demonstrated also that courses enabled learners to expand their social network, lessen their isolation, become more self-aware, reflect on themselves and their relationships with others, and reduce preconceptions regarding mental health [14,37,38,42]. 

These results from international studies concerned in-person RC programs that varied widely in length from a few hours to several days and afforded learners the possibility of attending courses more than once. The fact that we replicated these results with a shortened (three two-hour sessions for a total of six hours) online version is promising. They support the hypothesis that short online RC-type courses can slow the progression of psychological distress, prevent the deterioration of mental health in the general population and in at-risk groups, and break down social isolation in times of confinement measures and limited access to care (in addition to improving collective well-being and mutual compassion). Studies found that, during the COVID-19 crisis, anxiety and psychological distress levels shot up in several at-risk groups and in the general population [1,3,4,11]. This led individuals to isolate instead of seeking the help they needed. 

The circumstances of the COVID-19 pandemic undermined existing health care systems and shone a light on the limitations of a professionalized mental health system. Currently, mental health systems are based primarily on disease-focused biopsychomedical models, where solutions are handed down by health professions, and which depend on individual patient accountability. The pandemic has challenged this system, which has been unable to respond to an "overflow" of demands to access to care and services. Consequently, today more than ever, it is essential to invest in collective public health strategies for prevention and mental health promotion. The epidemiological data that have been amassed during and since the pandemic point to the importance of transforming our mental health systems and ensuring that personal skills are developed in individuals, living environments, and organizations in order to reduce the negative consequences of psychological distress and social isolation [43]. The step-care model for mental health disorders [44] emphasizes psychological health education and self-care as the first steps to intervene early to protect against mental health deterioration and prevent the onset or worsening of mental disorders. Health literacy is now recognized as an essential determinant of health to improve individual health outcomes, reduce inequities, and enhance service delivery. These interventions aim to improve people's knowledge and skills so that they regain control of their mental health, take action to protect it in their daily lives, and take a proactive role in their mental health.

Based on the perceptions of the learners interviewed, RC curriculum and courses addresses several social determinants of health:Improving mental health literacy (themes: “Acquiring new knowledge/Updating and expanding current knowledge”);Enhancing individual resilience and coping skills (themes: “Managing stress better and feeling more resilient and confident/Expressing yourself and becoming more confident”);Activating lifestyle habits, self-management strategies, and behaviors that support mental health (themes: “Using new tools daily to take care of you and your mental health/Becoming aware of the importance of taking care of yourself and your mental health”);Reducing prejudice, stigma, and epistemic inequities (themes: “Reflecting on and improving the way we deal with stigma/Recognizing the value of experiential knowledge”);Adopting attitudes and practices that support personal self-determination and make living together better (themes: “Modifying and refining your understanding of otherness/Becoming self-aware”).

These results also highlight the experience of socialization and co-learning between learners from diverse backgrounds (e.g., education and health care providers, university students, people in vulnerable situations, family/peer caregivers, workers, managers, retirees/seniors, citizens). Training sessions provide more than just knowledge and self-management strategies, they also afford learners the time and space to break their isolation, speak out, and be accepted into a group despite their mental health issues [18,19]. Participating in an RC course provides more than just an opportunity to gain knowledge, it also creates an opportunity for social interaction, mutual exchange, experimentation, and risk-taking. The stigma associated with mental illness leads to fear of acceptance, social isolation and exclusion, all of which RC courses help to combat by reducing stigma and promoting open dialogue and co-learning about mental health. This co-learning context where learners from diverse backgrounds discuss and learn together challenges the traditional model of intervention. Indeed, online courses allowed more accessibility (reduced costs, increased course variety and flexibility, etc.) to the learners. With only six hours of these, it was possible to engage in appropriate transformative learning and reflective processes. 

This qualitative study achieved its objectives. It helped us enrich our understanding of the impact of RC courses. This will allow us to formulate new hypotheses and enhance our ongoing experimental study by adding new measurement tools. Notably, an in-house tool has already been added to better identify what participants learn from the course (in terms of knowledge) and another is being developed to study ethical and interpersonal skills (social responsibility, and ethical and inclusive engagement) [45].

### Limits of the Study and Future Prospects

Despite the use of purposive sampling strategy (i.e., gender, age, courses attended, learner profile) to recruit learners with a wide range of profiles, the gender split was imbalanced, as only three of the fourteen participants were men. The results did not allow us to distinguish differences in perceived benefits by gender, age, learner profile, or course attended. Would a different gender distribution have provided a different perspective? Do younger learners (i.e., under 25) benefit differently from the training, given that their awareness of themselves and the world is still fledgling? Are the benefits different for mental health professionals with theoretical and clinical knowledge of mental health who are already equipped to deal with these issues? Do courses that directly address the ethical management of prejudice and stigmatization have a greater impact on attitudes and behaviors of openness toward others? It would take a different research design to be able to answer these questions.

The RC model involves bringing together people with different backgrounds to learn from one another. That is what makes it so powerful and appealing. However, to identify the specific benefits of RC courses with a view to finding better ways of meeting the needs of learners and at-risk groups would require stepping up the evaluation of the CASR education program in future. This entails increasing the number of interviews performed, conducting qualitative analyses by learner profile and course attended, and collecting quantitative data from as large a sample as possible in order to increase the statistical power of our ongoing experimental study. The availability of a large set of both quantitative and qualitative data will make triangulation possible, and this will lead to a better understanding of the impact of the shortened online version of RC courses and the interaction of different variables on outcomes.

The results of this study underscore the relevance of continuing research on the effectiveness of these online interventions on a global scale since they can provide easy access to mental health services and support. Several RCs around the world have adapted their curriculum and courses in response to the COVID-19 crisis but, to our knowledge, there is no other published study to date to have documented the effects of this format change on learners.

## 5. Conclusions

The RC courses delivered by the CASR, which applies the original model but was shortened and placed online in response to the constraints imposed by the COVID-19 pandemic, is a promising intervention for meeting current mental health needs. It can contribute to improve the mental health and self-determination of individuals in the general population and in at-risk groups. This can help reduce the social and economic burden currently associated with psychological distress and prevent the onset and worsening of mental disorders during public health crises. RC curriculum and courses foster the search for solutions outside the sphere of traditional health care facilities, in partnership with communities, in an intersectoral approach (education-health), thereby supporting the transformation of mental health care, services, and practices. By championing a public health approach centered on the social determinants of health and the creation of free and universally accessible learning environments devoid of prejudice and stigmatization, RCs act as a model of equity, diversity, and inclusion that promotes not only health but also better coexistence and interaction. This study contributes to new knowledge regarding mental health and the COVID-19 pandemic by showing that online RC courses benefit learners dealing with mental health issues. An RC education program is a way to get people to open up about psychological distress and mental health at the societal level, by helping create attitudes and behaviors of openness to others and mutual benevolence. 

## Figures and Tables

**Table 1 ijerph-20-02498-t001:** Purposive sampling strategy.

Criterion 1: Course Attended ^A,B^	Criterion 2: Gender ^C^	Criterion 3: Age Group (Years)	Criterion 4: Learner Profile ^D^
Course 1	F	20–30	University student
Course 2	F	30–40	Peer worker or person with experiential knowledge
Course 3	F	40–50	Health or education manager
Course 4	F	50–60	Health or education professional
Course 5	F	60+	Community member (caregiver or citizen)
Course 6	F	20–30	University student
Any	F	30–40	Peer worker or person with experiential knowledge
Course 1	M	40–50	Health or education manager
Course 2	M	50–60	Health or education professional
Course 3	M	60+	Community member (caregiver or citizen)
Course 4	M	20–30	University student
Course 5	M	30–40	Peer worker or person with experiential knowledge
Course 6	M	40–50	Health or education manager
Any	M	50–60	Health or education professional
Any	F/M	60+	Community member (caregiver or citizen)

^A^: Course 1: Increasing resilience, Course 2: Let’s talk about anxiety and worries, Course 3: Talking health, talking mental health, Course 4: Perspectives on stigma, Course 5: Finding meaning and motivation at work, Course 6: Toward well-being, dealing with stress. ^B^: Each course was six hours long, divided into three 2-h sessions. ^C^: F: Female, M: Male. ^D^: Each picked the main profile they identified with a time of course registration.

**Table 2 ijerph-20-02498-t002:** Participants’ characteristics (*n* = 14).

Participant	Gender	Age (Years)	Learner Profile	Course Attended
P1	M	44	Peer worker, person with experiential knowledge	Increasing resilience
P2	F	28	Citizen	Increasing resilience
P3	F	20	University student	Increasing resilience
P4	F	27	Health professional	Toward well-being, dealing with stress
P5	F	46	Health professional	Talking health, talking mental health
P6	F	66	Peer worker, person with experiential knowledge	Talking health, talking mental health
P7	M	61	Education professional	Finding meaning and motivation at work
P8	F	49	Health professional	Talking health, talking mental health
P9	M	61	Peer worker, person with experiential knowledge	Let’s talk about anxiety
P10	F	36	Education professional	Finding meaning and motivation at work
P11	F	38	Health professional	Perspectives on stigma
P12	F	65	Peer worker, person with experiential knowledge	Let’s talk about anxiety
P13	F	27	Health Manager	Perspectives on stigma
P14	F	44	Health Manager	Talking health, talking mental health

**Table 3 ijerph-20-02498-t003:** Categories and themes to emerge from the thematic analysis.

Categories	Themes
Updating and validating your mental health knowledge	Acquiring new knowledge
	Updating and expanding current knowledge
	Validating known and currently used strategies/tools/practices Continuing to integrate new information and training and wanting to learn
Taking care of yourself and your mental health	Using new tools daily to take care of you and your mental health
	Managing stress better and feeling more resilient and confident Becoming aware of the importance of taking care of yourself and your mental health Improving your motivation to work on yourself and to find help
Improving and modifying your behaviors and practices	Recognizing the value of experiential knowledge
	Reflecting on and improving the way we deal with stigma
Changing how you look at yourself and others	Becoming self-aware
	Modifying and refining your understanding of otherness
Interacting and connecting with others	Breaking isolation and feeling less lonely
	Expressing yourself and becoming more confident

## Data Availability

The datasets generated during the present study are available from the corresponding author, C.B., upon reasonable request. The datasets are not publicly available owing to privacy or ethical restrictions.

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
