# Peer review of "Learner Experience of an Online Co-Learning Model to Support Mental Health during the COVID-19 Pandemic: A Qualitative Study"

_ijerph, 2023, doi:10.3390/ijerph20032498_

Round 1
Reviewer 1 Report
>> The presentation of this manuscript very weak and its limited contributions
>> I think this paper is not innovative enough to meet the requirements of the journal.
Additional comments:
1. What is the main question addressed by the research?
Answer:
The aim of this study is to document the experience of learners who participated in online RC training during COVID-19 pandemic lockdown and their perceived impact on their mental health.
2. Do you consider the topic original or relevant in the field? Does it address a specific gap in the field?
Answer:
There is no research gab
3. What does it add to the subject area compared with other published material?
Answer:
It's added but the presentation very bad
4. What specific improvements should the authors consider regarding the methodology? What further controls should be considered?
Answer:
>> I feel that more explanation would be need on how the proposed method is performed.
>> If no one has proposed a method like the proposed algorithm, this claim should be highlighted much more. Else, it should be indicated who has done this, and it should be indicated what the innovations of the current paper are.
Author Response
I would like to thank the reviewers for their comments. The revised manuscript has been significantly improved to more accurately report the study. See table 1 attached for all details and point-by-point response.

Reviewer 2 Report
This paper shows the results about the experience of learners who participated in online RC training during COVID-19 pandemic lockdown and their perceived impact on their mental health, and I consider this topic quite useful and valuable.
However, some specific aspects of this manuscript need to be considered before the decision for publication. See below.
1.14 subjects' information should be placed in Data collection;
2. The Results are the focus of the article, so the author should present them in more detail. The author should tell the reader how the 14 themes and 5 category formed based on the interview content, because the process from the interview text to these topic and category is not completely direct, but through authors' processing.
3. At present, the discussion section of the manuscript is not enough. The author should draw more lessons from the existing research and theories to improve the depth of the discussion on the relationship between the experience of learners who participated in online RC training during COVID-19 pandemic lockdown and their mental health.
Author Response

(The authors gave the same response as above.)

Round 2
Reviewer 1 Report
The paper presents Learner Experience of an Online Co-learning Model to Support Mental Health During the COVID-19 Pandemic: A Qualitative Study. There are Some questions that need to addressed such as :
>> The language usage throughout this paper need to be improved, the author should do some proofreading on it. Give the article a mild language revision to get rid of few complex sentences that hinder readability and eradicate typo errors.
>> Your abstract does not highlight the specifics of your research or findings. Rewrite the Abstract section to be more meaningful. I suggest to be Problem, Aim, Methods, Results, and Conclusion.
>> Introduction section can be extended to add the issues in the context of the existing work and how proposed algorithms/approach can be used to overcome this.
>> The problems of this work are not clearly stated. There is ambiguity in statement understanding.
>> Add main contributions list as points in the Introduction section.
>> Add the rest organization section at the end of the Introduction section.
>> More clarifications and highlighted about the research gabs in the related works section.
>> Improve related works by discuss the recent studies such as:
- A Novel Low-Latency and Energy-Efficient Task Scheduling Framework for Internet of Medical Things in an Edge Fog Cloud System. Sensors
- Smart Healthcare System for Severity Prediction and Critical Tasks Management of COVID-19 Patients in IoT-Fog Computing Environments. Computational Intelligence and Neuroscience
- Blockchain multi-objective optimization approach-enabled secure and cost-efficient scheduling for the Internet of Medical Things (IoMT) in fog-cloud system. Soft Computing
- Bio-inspired robotics enabled schemes in blockchain-fog-cloud assisted IoMT environment. J. King Saud Univ. Comput. Inf. Sci
- Multi-Agent Systems in Fog–Cloud Computing for Critical Healthcare Task Management Model (CHTM) Used for ECG Monitoring. Sensors
- Mobile‐fog‐cloud assisted deep reinforcement learning and blockchain‐enable IoMT system for healthcare workflows. Transactions on Emerging Telecommunications Technologies
- Restricted Boltzmann machine Assisted Secure Serverless Edge System for Internet of Medical Things. IEEE Journal of Biomedical and Health Informatics
>> I feel that more explanation would be need on how the proposed method is performed.
>> Can you provide more information on the sample used in this study? How diverse was the sample in terms of age, gender, education level, etc.?
>> It would be interesting to see how the results of this study compare to other research on online mental health programs. Are there any notable similarities or differences?
>> How does this study contribute to the existing literature on mental health and the COVID-19 pandemic?
>> How can the findings of this study be used to improve the design and delivery of online mental health support programs?
>> This study is qualitative, would be interesting to see if these findings are consistent with a quantitative study. How generalizable are the findings to the population at large?
>> Can you provide more information on the Miles and Huberman’s stepwise content analysis method used in this study? How was it applied to the data?
>> It would be useful to see a more detailed description of each of the five themes that emerged. How were they defined and what specific examples were found within each theme?
>> How long was the duration of the study and did participants have to complete the training courses to be interviewed?
>> Were there any limitations to the study that you would like to highlight?
>> Results need more explanations. Additional analysis is required at each experiment to show the its main purpose.
>> The Limitations of the proposed study need to be discussed before conclusion.
>> The conclusion of the paper and the implications of the work, whether it is the conclusion of the research or the conclusion of the manuscript.
